

# Genome-wide analysis of growth-regulating factors (GRFs) in *Triticum aestivum*

Wendi Huang[1], Yiqin He[1], Lei Yang[1], Chen Lu[1], Yongxing Zhu[1], Cai Sun[2], Dongfang Ma[1,3] and Junliang Yin[1,3]

[1] Engineering Research Center of Ecology and Agricultural Use of Wetland, Ministry of Education/Hubei Collaborative Innocation Center for Grain Industry/College of Agriculture, Yangtze University, Jingzhou, Hubei, China

[2] Plant Protection and Fruiter Technical Extension Station, Wanzhou District, Chongqing, China

[3] Ministry of Agriculture Key Laboratory of Integrated Pest Management in Crops in Central China, Institute of Plant Protection and Soil Science, Hubei Academy of Agricultural Sciences, Wuhan, Hubei, China

## ABSTRACT

The Growth-Regulating Factor (GRF) family encodes a type of plant-specific transcription factor (TF). GRF members play vital roles in plant development and stress response. Although GRF family genes have been investigated in a variety of plants, they remain largely unstudied in bread wheat (*Triticum aestivum* L.). The present study was conducted to comprehensively identify and characterize the *T. aestivum* GRF (*TaGRF*) gene family members. We identified 30 *TaGRF* genes, which were divided into four groups based on phylogenetic relationship. TaGRF members within the same subgroup shared similar motif composition and gene structure. Synteny analysis suggested that duplication was the dominant reason for family member expansion. Expression pattern profiling showed that most *TaGRF* genes were highly expressed in growing tissues, including shoot tip meristems, stigmas and ovaries, suggesting their key roles in wheat growth and development. Further qRT-PCR analysis revealed that all 14 tested *TaGRFs* were significantly differentially expressed in responding to drought or salt stresses, implying their additional involvement in stress tolerance of wheat. Our research lays a foundation for functional determination of TaGRFs, and will help to promote further scrutiny of their regulatory network in wheat development and stress response.

## INTRODUCTION

Growth-Regulating Factors (GRFs) are plant-specific transcription factors that play important roles in regulating plant growth and abiotic stress response (*Kim et al., 2012*; *Baucher et al., 2013*). The first GRF gene *OsGRF1* was identified from rice, where it was shown to play an essential role in regulating gibberellic acid (GA)-induced stem elongation (*Van der Knaap, Kim & Kende, 2000*). In recent years, with the development of reference genomes, many GRF genes have been identified and characterized from plant species at genome-wide levels (*Omidbakhshfard et al., 2015*). Protein sequence analysis has determined that there are two conserved domains, QLQ (Gln, Leu, Gln) and WRC (Trp,

Corresponding author
Dongfang Ma, madf@yangtzeu.edu.cn

Arg, Cys), in the N-terminal region of the GRF protein (*Kim, Choi & Kende, 2003*). The QLQ domain serves as a protein-protein interaction feature which can interact with the GRF-interacting factor (GIF) (*Kim & Kende, 2004*). The WRC domain is mainly involved in DNA binding and consists of a functional nuclear localization signal and a DNA binding motif (zinc finger structure) (*Choi, Kim & Kende, 2004*). Unlike the conserved amino acid residues in the N-terminal region, the C-terminal region of GRF is variable, with some studies demonstrating that the C-terminal region has trans-activation activity (*Choi, Kim & Kende, 2004*; *Kim & Kende, 2004*; *Liu et al., 2014a*; *Liu et al., 2014b*). In addition, the C-terminal region may contain several low conservative motifs, such as TQL (Thr, Gln, Leu) and FFD (Phe, Phe, Asp) (*Zhang et al., 2008*).

Currently, the GRF transcription factors have been reported in *Arabidopsis* (*Kim, Choi & Kende, 2003*), rice (*Choi, Kim & Kende, 2004*), maize (*Zhang et al., 2008*), Chinese cabbage (*Wang et al., 2014*), soybean (*Chen et al., 2019*) and tea (*Wu, Wang & Zhuang, 2017*). In these plants, GRF genes are strongly expressed in tissues involved in active growth and development, such as stem tips, flower buds, and immature leaves, but weakly expressed in mature tissues or organs. They can participate in the early growth and development of plants and play an important regulatory role in the formation of plant tissues or organs, such as leaf development, stem elongation and root growth (*Bazin et al., 2013*; *Kuijt et al., 2014*; *Wu et al., 2014*). *GRF* genes have been reported as positive regulators of leaf size by promoting and/or maintaining the proliferation activity of leaf primordia cells (*Horiguchi, Kim & Tsukaya, 2005*; *Kim & Lee, 2006*). For example, overexpression of *AtGRF1*, *AtGRF2*, and *AtGRF5* resulted in larger than normal leaves in wild-type (WT) *Arabidopsis*, while the leaves of *grf* mutants, such as *grf3 - 1*, *grf5 - 1*, *grf1 - 1/grf2*, *grf2/grf3*, and *grf1/2/3*, were much smaller than the WT (*Debernardi et al., 2014*; *Horiguchi, Kim & Tsukaya, 2005*). *GRF2* was found to enhance seed oil production in rapeseed (*Brassica napus*) by regulating cell number and plant photosynthesis (*Liu et al., 2012*). GRF TFs not only participate in plant growth and development, but also respond to certain abiotic stresses (*Kim et al., 2012*). In *Arabidopsis*, while under stress conditions *AtGRF7* expression is inhibited to activate osmotic stress-responsive genes (*Kim et al., 2012*). Functional classification of the *AtGRF1* and *AtGRF3* downstream genes suggests that most target genes are involved in defense responses and disease resistance processes (*Liu et al., 2014a*; *Liu et al., 2014b*).

Although bread wheat is one of the world's most important food crops, accounting for more than half of total human consumption (*Ma et al., 2016*; *Sun et al., 2019*; *Yin et al., 2018a*), its production is seriously threatened by biotic and abiotic stress factors, including drought, salinity, and extreme temperatures (*Yin et al., 2019*; *Zhu et al., 2015*; *Zhu et al., 2019a*). Although genome-wide analyses of GRF transcription factors have been performed to a number of plant species, including *Arabidopsis* (*Kim, Choi & Kende, 2003*), rice (*Choi, Kim & Kende, 2004*), maize (*Zhang et al., 2008*), Chinese cabbage (*Wang et al., 2014*), soybean (*Chen et al., 2019*) and tea (*Wu, Wang & Zhuang, 2017*), genome-wide identification and characterization have not yet been conducted to common wheat GRF (TaGRF) family members.

In this study, bioinformatics methods were used to systematically analyze the TaGRF TFs, including sequence characteristics, chromosome distribution, phylogenetic relationship,
gene structure, and conserved motif and domain prediction. In total, 30 TaGRFs were identified from the wheat genome. On this basis, the gene expression patterns of wheat GRF were analyzed based on RNA-seq data from different wheat tissues. The expression patterns of *TaGRFs* under drought and salt stresses were also analyzed by qRT-PCR.

## MATERIAL AND METHODS

### Identification of GRF genes in *T. aestivum*, *T. urartu*, *T. dicoccoides* and *Ae. tauschii*

Genome-wide data for *Triticum aestivum* (IWGSC v1.1), *Triticum urartu* (v1.43), *Triticum dicoccum* (v1.0.43), and *Aegilops tauschii* (v4.0.43) were downloaded from Ensembl Plants database (http://plants.ensembl.org/index.html). First, the Hidden Markov Model (HMM) of WRC (PF08879) and QLQ (PF08880) domains were obtained from PFAM (http://pfam.xfam.org/) and used as query sequences for HMMER3.0 (http://hmmer.org/download.html) searching (*e*-value $\leq 1e^{-10}$). Second, download of known GRF protein sequence were used as query sequences, including 9 GRFs from *Arabidopsis* (*Berardini et al., 2015*), 14 GRFs from *Zea mays* (*Andorf et al., 2015*), and 12 GRFs from *Oryza sativa* (*Ouyang et al., 2006*). They were then used as query sequences for BLASTp searching the wheat database. The first uncurated protein sequence list was genereted by e-values lower than $1 \times 10^{-10}$. Next, we combined the results and deleted the redundant sequences. Finally, predicted proteins were considered as GRFs only if they contained QLQ and WRC conserved domains verified by NCBI CDD (https://www.ncbi.nlm.nih.gov/Structure/cdd/wrpsb.cgi) and SMART (http://smart.embl-heidelberg.de/) (*Letunic & Bork, 2017*).

### Characterization of TaGRFs proteins

ExPASy server10 (https://web.expasy.org/compute_pi/) was used to predict the amino acid length, molecular weight (MW), isoelectric point (pI), stability, and grand average of hydropathicity (GRAVY) for TaGRFs proteins (*Li et al., 2018*); and subcellular localization prediction was carried out by Plant-mPLoc (http://www.csbio.sjtu.edu.cn/bioinf/plant-multi/) (*Chou & Shen, 2010*).

### Chromosomal location and gene duplication of TaGRFs

The wheat genome GFF3 gene annotation file was from the wheat database IWGSC v1.1 (https://wheat-urgi.versailles.inra.fr/Seq-Repository/Assemblies). Gene structure annotations of *TaGRFs* were extracted from the GFF3 file. The start and end location information of the *TaGRFs* in the corresponding chromosomes was used to draw the physical map by the software MapInspect (*Fang et al., 2019*). The orthologous genes from wheat and its subgenome donor were identified by the common tool "all against all BLAST search". The cutoff values (e-value $<10^{-10}$, identity $>80\%$) were used to assure the reliability of the orthologues. Then we used Multiple Collinearity Scan toolkit (MCScanX) to depict their homology relationships (*Wang et al., 2012*). Synteny diagrams were generated using the R package "circlize". Gene duplication events were divided into tandem duplication events and segmental duplication events. Tandem duplication events were determined
by the following evaluation criteria: (1) length of the aligned sequence >80% but of each sequence, (2) identity >80%, (3) threshold $\leq 10^{-10}$, (4) only one duplication can be recognised when genes are tightly linked; and (5) intergenic distance is less than 25 kb. When genes passed the criteria for (1), (2), and (3), but were on a different chromosome, they were deemed to be segmental duplications (*Fang et al., 2020*; *Jiang et al., 2020*).

## Analysis of TaGRFs gene structures and motifs

According to the *TaGRFs* annotation information, GSDS2.0 (http://gsds.gao-lab.org/Gsds_about.php) was used to produce *TaGRFs* genetic structure (*Hu et al., 2017*). The MEME v4.9.1 (http://meme-suite.org/) was used to identify conserved TaGRF protein motifs (*Zheng et al., 2017*). The trained parameters were applied as follows: each sequence may contain any number of nonoverlapping occurrences of each motif, up to 20 different motifs, and a motif width range of 6 to 50 amino acids (aa). These motif patterns were drawn using TBtools software (*Chen et al., 2020*). The annotations of those predicted motifs were analyzed by SMART (http://coot.embl-heidelberg.de/SMART/) (*Letunic & Bork, 2017*). Multiple amino acid sequences were aligned using DNAMAN6.0 (Lynnon Biosoft).

## Phylogenetic analyses of TaGRFs

The 99 protein sequences (9 AtGRFs, 12 OsGRFs, 14 ZmGRFs, 30 TaGRFs, 6 TuGRFs, 10 AeGRFs, and 18 TdGRFs) were conducted multiple comparisons by using ClustalW2 software (*Thompson, Higgins & Gibson, 1994*). Then, the phylogenetic relationships were inferred using the Neighbor-Joining (NJ) method with bootstrap analysis for 1,000 repetitions by MEGA7.0 (*Kumar, Stecher & Tamura, 2016*). Finally, the midpoint rooted base tree was drawn using Interactive Tree of Life (IToL, v4, http://itol.embl.de) (*Letunic & Bork, 2019*).

## *Cis*-acting elements analysis of TaGRFs

The PlantCARE (http://bioinformatics.psb.ugent.be/webtools/plantcare/html/) was used to predict *cis*-acting elements in the regions 1,500 bp upstream of 30 *TaGRFs* start codons (*Lescot et al., 2002*). The predicted results were organized and displayed by the R package "pheatmap" (*Jiang et al., 2019*; *Zhu et al., 2019b*).

## Gene Ontology annotation in TaGRF family genes

The functional annotation of GRF sequences and the analysis of annotation data were performed using Blast2GO (http://www.blast2go.com) (*Conesa et al., 2005*). The full-length amino acid sequence of the TaGRF proteins were uploaded to the original program, drawn and annotated. The program provides the output defining three categories of GO classification namely biological processes, cellular components, and molecular functions.

## Multiple conditional transcriptome analysis of TaGRFs

The multiple transcriptome data were downloaded from the Wheat Expression Browser (http://www.wheat-expression.com/) (*Ramírez-González et al., 2018*); and the heat maps of *TaGRFs* were generated using the R package "pheatmap".
## Growth and stress treatment of wheat seedlings

Seeds of Emai 170 (a hexaploid common wheat cultivars) were sterilized on the surface with 1% hydrogen peroxide, rinsed thoroughly with distilled water, and germinated in an incubator at 25 °C for 2 days (*He et al., 2020*; *Ma et al., 2016*). According to the reported method, the seedlings were transferred to 1/2 strength Hoagland nutrient solution and cultured in continuous ventilation (*Yin et al., 2019*; *Zhu, Gong & Yin, 2019*). After five days (when wheat seeding reached the stage of one heart and one leaf), 85.5 mM NaCl and 82.5 mM mannitol were applied to seedings. Every two days, 2 M KOH or 0.4 M $H_2SO_4$ was used to adjust the pH of culture solution to 6.0. During the application, the plants were grown at 16 h/8 h (day/night) and 25 °C. Leaves and roots were collected at 2 h, 4 h, 8 h, 12 h, 24 h, 96 h and 144 h after treatments. Three biological repeats are included for each treatment. Finally, the samples were immediately frozen with liquid nitrogen and stored at −80 °C.

## RNA isolation and qRT-PCR analysis

According to the manufacturer's instructions, total RNA of samples were extracted by TRizol reagent (Invitrogen, USA) and cleansing DNA with DNaseI (TaKaRa, USA). The first cDNA was reverse-transcribed from RNA by RevertAid Reverse Transcriptase (Vazyme, China). Gene-specific primers were designed using Primer 5.0; and the ADP-ribosylation factor *Ta2291* (F: GCTCTCCAACAACATTGCCAAC, R: GCTTCTGCCTGTCACATACGC) was used as an internal reference gene for qRT-PCR analysis (*Paolacci et al., 2009*). The qRT-PCR reaction system and protocol were carried out as manufacturer's instructions for SYBR® (Vazyme, China). For each sample, settings included three technical replicates. Relative gene expression level was calculated using the $2^{-\Delta\Delta Ct}$ method (*Yin et al., 2018b*).

# RESULTS

## Identification and analysis of wheat GRF transcription factor gene family members

For identification of GRF TF genes in wheat, both BLAST and Hidden Markov Model (HMM) searches were performed. The 35 known GRF proteins (Table S1), including *Arabidopsis* (9), maize (14), rice (12) as the query sequences to conduct BLASTp against the wheat reference genome IWGSCv1.1. Using the HMM of WRC (PF08879) and QLQ (PF08880) domains were used as the query sequences for HMMER3.0 searching. The candidate proteins were verified by NCBI CDD and SMART Online Tools to determine that the TaGRF contained both WRC and QLQ domains. Finally, a total of 30 TaGRFs were identified from the wheat genome. We named wheat GRF genes (*TaGRFs*) according to the naming rule of *Schilling et al. (2020)*;  the corresponding gene IDs are shown in Table 1. Using the same method, we identified 6, 10, and 18 GRFs from *T. urartu*, *Ae. tauschii* and *T. dicoccoides*, respectively (Table S2). The deduced polypeptides ranged in length from 206 (TaGRF2-2A) to 611 (TaGRF4-4B) amino acids, with the predicted molecular weights ranging between 21.6 to 64.2 kDa. Their isoelectric points ranged from 4.72 (TaGRF1-2B) to 10.23 (TaGRF2-2A). Their instability parameters were between 41.46

**Table 1  Protein features of GRFs in *Triticum aestivum*.**

| Name | Locus ID | Len | MW | PI | II | stability | GRAVY | Sub |
|---|---|---|---|---|---|---|---|---|
| TaGRF1-2A | TraesCS2A02G238700.1 | 319 | 34683.79 | 4.89 | 47.17 | unstable | −0.492 | Chloroplast. Cytoplasm. Nucleus. |
| TaGRF1-2B | TraesCS2B02G256600.1 | 258 | 27746.68 | 4.72 | 54.92 | unstable | −0.736 | Nucleus. |
| TaGRF1-2D | TraesCS2D02G246600.1 | 264 | 28182.2 | 4.76 | 55.93 | unstable | −0.692 | Nucleus. |
| TaGRF2-2A | TraesCS2A02G398300.1 | 206 | 21620.58 | 10.23 | 49.79 | unstable | −0.269 | Chloroplast. Nucleus. |
| TaGRF2-2B | TraesCS2B02G416300.1 | 227 | 24077.17 | 9.82 | 53.68 | unstable | −0.479 | Nucleus. |
| TaGRF2-2D | TraesCS2D02G395900.1 | 229 | 24221.3 | 9.57 | 49.91 | unstable | −0.443 | Nucleus. |
| TaGRF3-2A | TraesCS2A02G435100.1 | 384 | 42326.85 | 6.76 | 61.22 | unstable | −0.623 | Nucleus. |
| TaGRF3-2B | TraesCS2B02G458400.1 | 387 | 42454.97 | 7.01 | 61.92 | unstable | −0.621 | Nucleus. |
| TaGRF3-2D | TraesCS2D02G435200.1 | 391 | 42780.33 | 7.04 | 60.85 | unstable | −0.62 | Nucleus. |
| TaGRF4-4A | TraesCS4A02G255000.1 | 607 | 63953.28 | 6.87 | 51.64 | unstable | −0.416 | Nucleus. |
| TaGRF4-4B | TraesCS4B02G060000.1 | 611 | 64277.53 | 6.72 | 51.31 | unstable | −0.423 | Nucleus. |
| TaGRF4-4D | TraesCS4D02G059600.1 | 578 | 61162.1 | 6.58 | 53.81 | unstable | −0.45 | Nucleus. |
| TaGRF5-4A | TraesCS4A02G291500.1 | 408 | 45327.7 | 9 | 61.39 | unstable | −0.842 | Nucleus. |
| TaGRF5-4D | TraesCS4D02G020300.1 | 415 | 45994.94 | 8.82 | 62.06 | unstable | −0.794 | Nucleus. |
| TaGRF6-4A | TraesCS4A02G434900.1 | 371 | 39941.33 | 8.5 | 61.23 | unstable | −0.726 | Nucleus. |
| TaGRF7-6A | TraesCS6A02G174800.1 | 315 | 33604.85 | 8.12 | 41.46 | unstable | −0.35 | Nucleus. |
| TaGRF8-6A | TraesCS6A02G257600.1 | 212 | 22597.56 | 9.54 | 52.41 | unstable | −0.384 | Nucleus. |
| TaGRF8-6B | TraesCS6B02G267500.1 | 211 | 22347.22 | 9.64 | 53.2 | unstable | −0.382 | Nucleus. |
| TaGRF8-6D | TraesCS6D02G238900.1 | 215 | 22750.77 | 9.9 | 55.13 | unstable | −0.36 | Nucleus. |
| TaGRF9-6A | TraesCS6A02G269600.1 | 408 | 43457.29 | 7.65 | 65.49 | unstable | −0.579 | Nucleus. |
| TaGRF9-6B | TraesCS6B02G296900.1 | 406 | 43435.28 | 8.46 | 64.46 | unstable | −0.571 | Nucleus. |
| TaGRF9-6D | TraesCS6D02G245300.1 | 409 | 43620.48 | 8.16 | 64.18 | unstable | −0.559 | Nucleus. |
| TaGRF10-6A | TraesCS6A02G335900.1 | 409 | 44786.45 | 7.22 | 51.42 | unstable | −0.833 | Nucleus. |
| TaGRF10-6B | TraesCS6B02G366700.1 | 410 | 44724.4 | 7.21 | 51.02 | unstable | −0.821 | Nucleus. |
| TaGRF10-6D | TraesCS6D02G315700.1 | 414 | 45263.95 | 7.24 | 51.94 | unstable | −0.835 | Nucleus. |
| TaGRF11-7A | TraesCS7A02G049100.1 | 370 | 40161.6 | 8.78 | 59.57 | unstable | −0.762 | Nucleus. |
| TaGRF11-7D | TraesCS7D02G044200.1 | 368 | 39895.24 | 8.57 | 58.69 | unstable | −0.75 | Nucleus. |
| TaGRF12-7A | TraesCS7A02G165600.1 | 309 | 34214.05 | 8.55 | 65.62 | unstable | −0.841 | Nucleus. |
| TaGRF12-7B | TraesCS7B02G070200.1 | 316 | 34886.68 | 8.55 | 66.25 | unstable | −0.882 | Nucleus. |
| TaGRF12-7D | TraesCS7D02G166400.1 | 320 | 35405.24 | 8.26 | 63.68 | unstable | −0.882 | Nucleus. |

**Notes.**

Len, Lengths (aa); MW, molecular weight (kD); pI, Isoelectric point; II, instability index; GRAVY, Grand average of hydropathicit; Sub, Subcellular localization.

(TaGRF7-6A) to 66.25 (TaGRF12-7B). Their average hydrophilicity coefficient ranged from 0.269 (TaGRF2-2A) to 0.882 (TaGRF12-7B) (Table 1). Subcellular localization predictions showed that all GRF proteins except TaGRF1-2A and TaGRF2-2A were localized only in the nucleus, while TaGRF1-2A was located in the chloroplast, cytoplasm and nucleus, and TaGRF2-2A was located in the chloroplast and nucleus.

According to the phylogenetic relationships (Fig. 1), the 30 TaGRFs could be divided into four sub-categories (Group I to IV). Group I consisted of a single member, *TaGRF7-6A*. Group II included *TaGRF1-2A*, *TaGRF2-2A*, *TaGRF1-2B*, *TaGRF2-2B*, *TaGRF1-2D*,
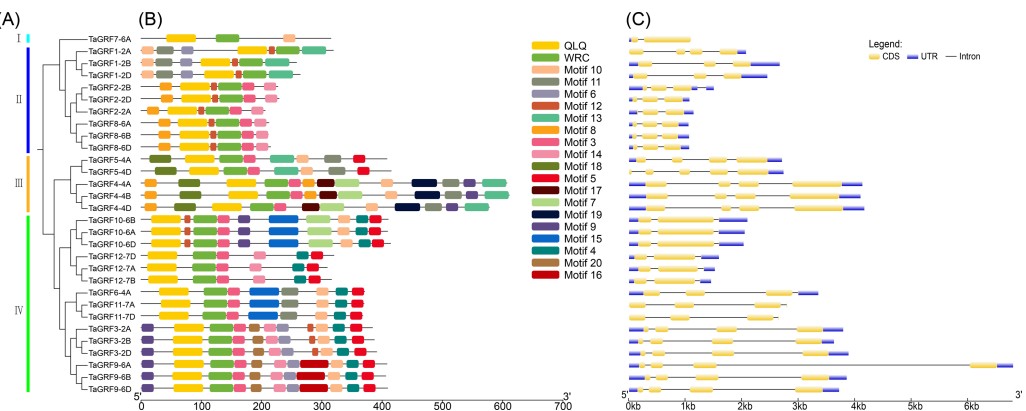

**Figure 1** **Gene structure and motif analysis of *TaGRF*.** (A) The phylogenetic tree of TaGRF. This tree consists of 1,000 bootstraps created by the Neighbor-Joining (NJ) method in MEGA7. (B) The motif of TaGRF was identified by MEME. MAST was used to display patterns. Each pattern is represented by a specific color. A red dot indicates a motif associated with a functional domain. (C) Exon-intron structure of *TaGRF*. Exon-intron structure analysis was performed using GSDS. The length of exons and introns is shown proportionally. Uuntranslated regions (UTR) are represented by blue boxes, exons are indicated by yellow boxes, and introns are indicated by black lines.

*TaGRF2-2D*, *TaGRF8-6A*, *TaGRF8-6B*, and *TaGRF8-6D*. Group III consisted of *TaGRF4-4A*, *TaGRF5-4A*, *TaGRF4-4B*, *TaGRF5-4D*, and *TaGRF4-4D*. The remaining *TaGRF* genes were classified in Group IV.

The *TaGRF* gene structure map showed that all wheat GRF gene members contain 1 to 4 introns, with the majority having 2 to 3 introns (Fig. 1, and Table S3). There were 2 to 5 exons, with most *TaGRF* having 2 to 4 exons. The exon number of *TaGRF* genes within same group were relatively consistent.

Conservative motif analysis indicated that TaGRF protein domains are highly conserved among the 30 members. Each member contains only two structural domains: WRC (Motif 1) and QLQ (Motif 2) (Fig. 1). Lengths and the most matching sequences of 20 motifs were shown in Table S4.

In order to further analyze the conservation degree of QLQ and WRC domains in TaGRFs, we performed multiple sequence alignment of these two domains. The results indicate that, as highlighted in Fig. 1, the QLQ and WRC motifs are highly conserved. The N-terminal QLQ motif was conserved with one Leu and two Gln in all the TaGRF proteins. The WRC motif was also highly conserved with one Trp, Arg, and Cys in each of the TaGRF proteins. A zinc finger motif (CCCH) was also found within the WRC domain in all TaGRF proteins (Fig. 2).

## Chromosome localization of wheat *TaGRF* genes

Based on the GFF3 genome reference files, the chromosome map of *TaGRF* genes was generated using MapInspect software (Fig. 3, and Table S3). The three sub-genomes A, B, and D contained 15, 11, and 10 *TaGRFs*, respectively. But the *TaGRFs* are not uniformly distributed among chromosomes (chromosome 2, 9; chromosome 4, 6; chromosome 6,

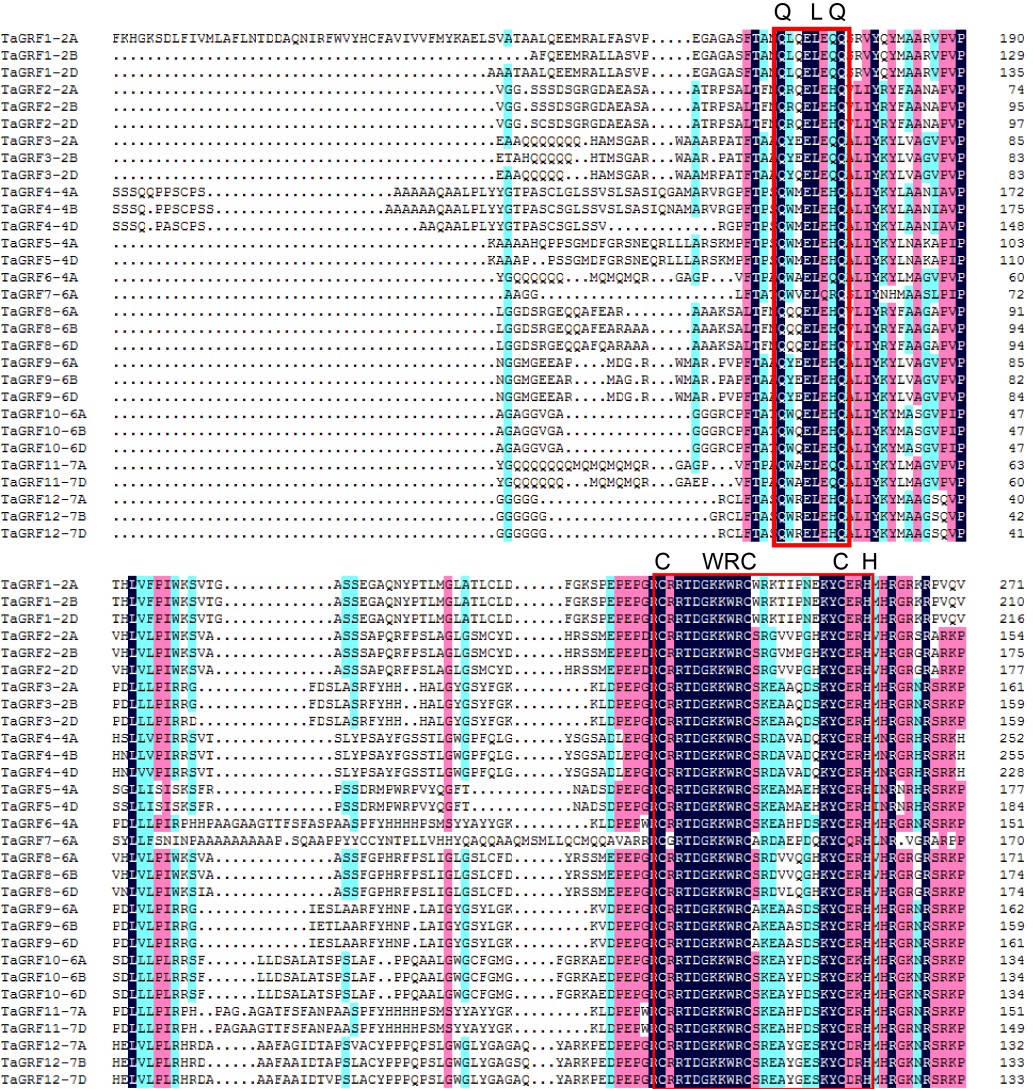

**Figure 2** **Protein sequence alignment of TaGRFs.** The functional areas are indicated by red boxes.

10; and chromosome 7, 5). However, distribution range of genes in different group was diverse. Members of *TAGRF* group II are only distributed on chromosome 4.

## Phylogenetic analysis of GRF transcription factor family members in wheat, rice, maize, *Arabidopsis, T. urartu*, *Ae. tauschii* and *T. dicoccoides*

The phylogenetic analysis of wheat (30), rice (12), maize (14), *Arabidopsis* (9), *T. urartu* (6), *Ae. tauschii* (10) and *T. dicoccoides* (18) GRFs showed that 99 GRFs could be divided into 4 sub-categories (Group I to IV) (Fig. 4). Group I contained only *AtGRF7* and *AtGRF8* of *Arabidopsis*, *TaGRF7-6A* of wheat. The Group II included 9 TaGRFs, 3 AtGRFs, 2 ZmGRFs, 3 OsGRFs, 3 AeGRFS, 6 TdGRFs, and 3 TuGRFs. The group III consisted of 5 TaGRFs, 2 AtGRFs, 3 ZmGRFs, 4 OsGRFs, 2 AeGRFs, and 4 TdGRFs. The Group IV included,

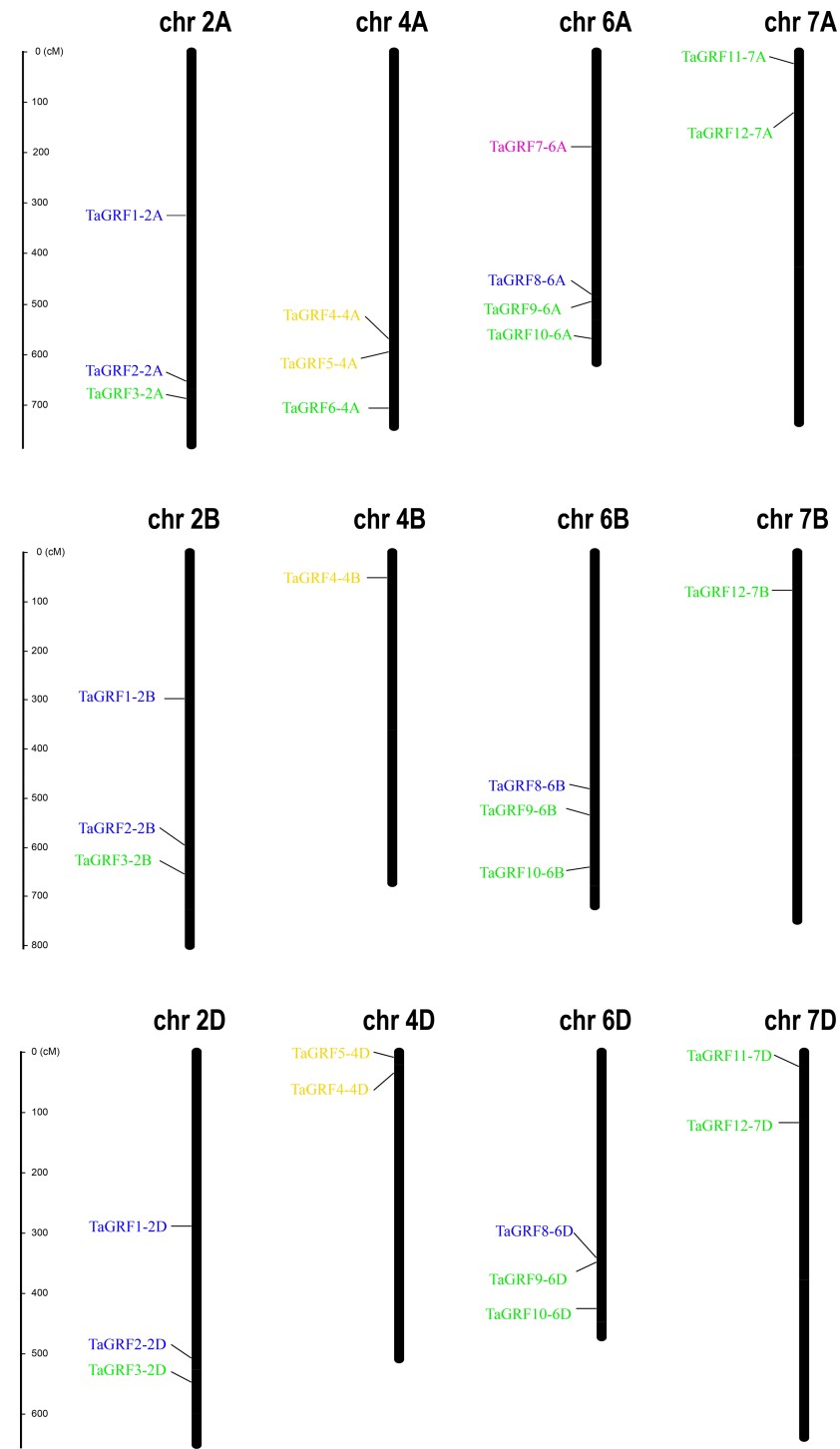

**Figure 3** **Chromosome locations of the 30 *TaGRFs* in wheat.** Different sub-groups of *TaGRFs* are represented in different colors: purple, Group I; blue, Group II; yellow, Group III; green, Group IV. In addition. Chr, Chromosome. The starting and ending information for the 30 chromosomal *TaGRFs* are listed in Table S2.

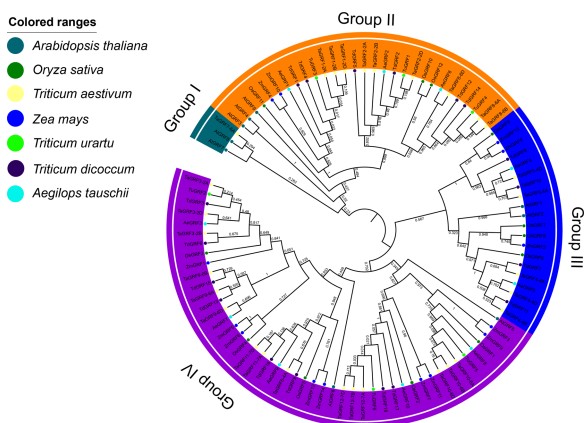

**Figure 4** **Phylogenetic tree of GRFs predicted in bread wheat and known in maize, rice, *Arabidopsis*, *T. urartu*, *T. dicoccoides* and *Ae. tauschii*.** All amino acid sequences were aligned using ClustalW. The phylogenetic tree was constructed by the Neighbor-Joining (1,000 replicates) method using MEGA7.0. Different groups are distinguished by different colored ribbons. GRFs from wheat, maize, rice, *Arabidopsis*, *T. urartu*, *T. dicoccoides* and *Ae. tauschii* are distinguished with different colored circles.

15 TaGRFs, 2 AtGRFs, 5 OsGRFs, 9 ZmGRFs, 5 AeGRFs, 8 TdGRFs, and 3 TuGRFs. In addition, we found that in Group II - IV, TaGRF has a closer phylogenetic relationship with TuGRF, AeGRF and TdGRF, followed by OsGRF and ZmGRF. and relatively distant from the AtGRF.

### *TaGRF* gene promoter *Cis-* element analysis

Analysis of the *cis*-elements in the promoter sequence was important for understanding the regulatory functions of genes. The *cis*-acting element analysis was performed in Plant-CARE by using upstream sequences (1.5 kb) of *TaGRF* genes extracted from the wheat genome. The detailed information including function and location were displayed in Table S5. The results showed that all 30 *TaGRF* genes contained several TATA boxes and CAAT boxes, indicating that TaGRF genes can be normally transcribed. When focusing on the *cis*-acting elements associated with wheat growth and development, hormonal and stress responses, it can be seen in Fig. 5 the *TaGRF* gene promoter contains a large number of *cis*-elements, with the largest number found in *TaGRF9-6D* having 19 *cis*-elements, and *TaGRF3-2A* with the least, containing only 8 *cis*-elements. There were several different light-related elements in these *cis*-elements, such as AE-box, Box 4, I-box, C-box, Sp1, circadian, CAG-motif, 3-AF1 binding site, LAMP-element, TCT-motif, GATT-motif, ATCT-motif, and Gap-box. This suggests that the GRF gene family may play a role in light response.

In addition, a large number of responsive hormones and stress-related *cis*-elements were found in the promoter region of the *TaGRFs*, including auxin (11 TGA-elements), gibberellin (8 GARE-motifs and 6 P-boxes), jasmonic acid methyl ester (65 TGACG-motifs), abscisic acid (72 ABREs) and other hormone response components, as well as anaerobic induction (25 AREs), drought (19 MBSs) and low temperature (3 LTRs) and other stress response *cis*-elements. This suggests a potential role for the wheat GRF family in wheat growth and development and in a variety of hormones and stress.
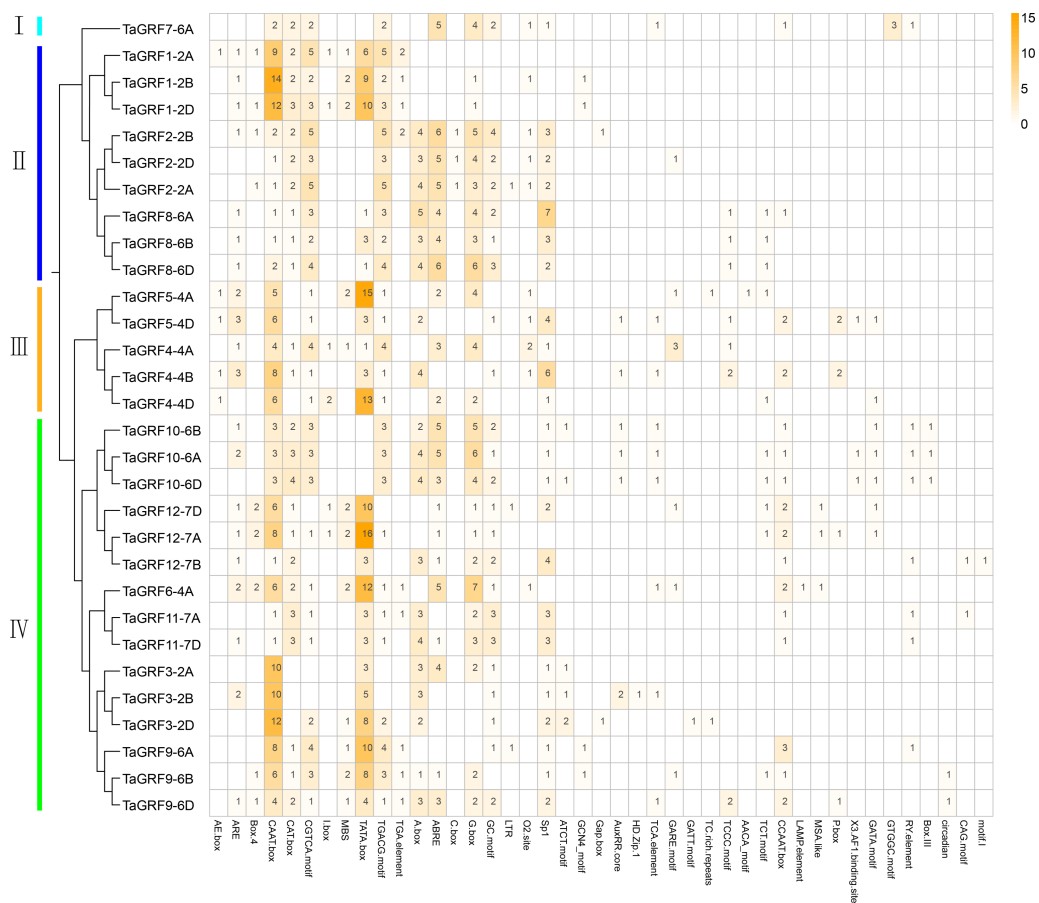

**Figure 5** The *cis*-acting element involved in stress responses of the *TaGRF* genes promoters. Different colors and numbers on the grid indicate numbers of different promoter elements in each TaGRF gene. All elements in the *TaGRF* gene promoter are listed in Table S5.

## Gene Ontology annotation in TaGRF family genes

The GO item analysis was performed using Blast2Go and the results indicated the putative participation of 30 TaGRF proteins in diverse biological processes (Fig. 6, and Table S6). Total ten different GO items of biological processes were defined. Majority of the TaGRFs were predicted to function in 'regulation of transcription, DNA-templated (GO: 0006355)' (76.67%), followed by 'response to deep water (GO: 0030912)' (20%) and 'response to gibberellin (GO: 0009739)' (20%). Molecular function prediction showed that about 76.67% of the TaGRFs were evidenced to participation of 'ATP binding (GO: 0005524)' and 'hydrolase activity, acting on acid anhydrides, in phosphorus-containing anhydrides (GO: 0016818)'. Cellular localization prediction indicated that the majority of TaGRF proteins (80%) were localized in the nucleus (Fig. 6).

## Homologous gene pairs and synteny analysis

Gramineae evolved 50–70 million years ago, and the Pooideae subfamily, which includes barley and wheat, evolved about 20 million years ago (*Inda et al., 2008*;

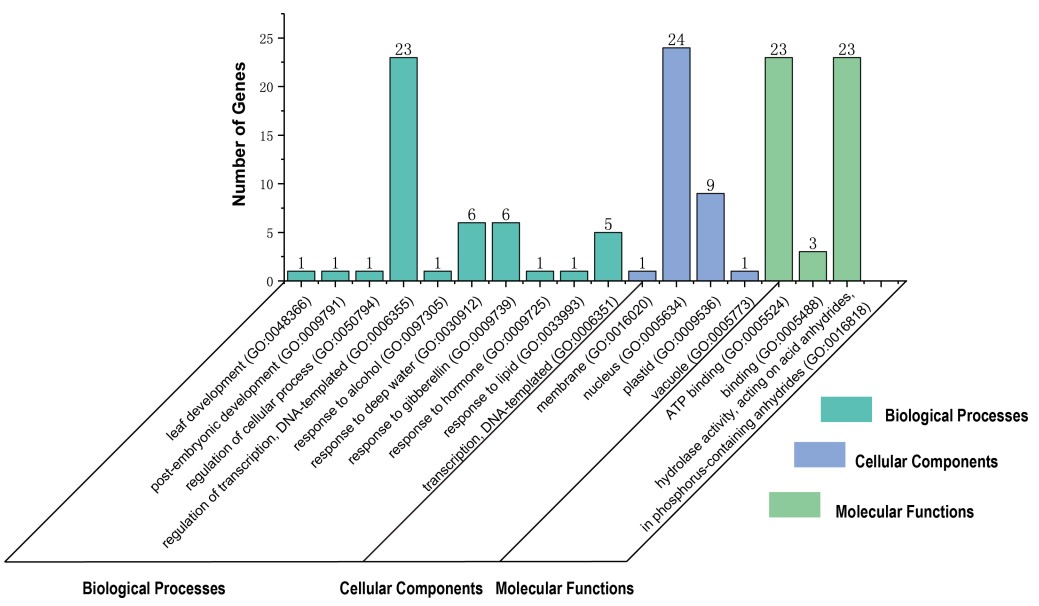

**Figure 6  Gene Ontology (GO) distributions for the TaGRF proteins.** The gene ontology under three categories, biological processes, molecular functions and cellular component.

*Peng, Sun & Nevo, 2011*). Obviously, common wheat has a complicated evolutionary history, and its ancestors' origin is affected by many factors, but research shows that wheat has two major polyploid evolutionary events (*Ling et al., 2013*). Homology reflects the phylogeny of a species, so it can be used to transfer annotations for one known gene to another newly sequenced genome. In order to further infer the evolutionary origin and homology of the wheat GRF family, Sixty-four *GRFs* were identified from *T. aestivum* (30 *TaGRFs*), *T. urartu* (6 *TuGRFs*), *T. dicoccoides* (18 *TdGRFs*) and *Ae. tauschii* (10 *AeGRFs*) using a computer-based method (Fig. 7A, and Table S7). There were no paralogous gene pairs in *Ae. tauschii* and *T. urartu*, and 21 and 5 paralogous gene pairs in *T. aestivum* and *T. dicoccoides*, respectively. Among them, 18 orthologous gene pairs were identified from *T. aestivum* and *Ae. tauschii*, 5 orthologous gene pairs were identified from *T. dicoccoides* and *Ae. tauschii*, 5 orthologous gene pairs were identified from *T. urartu* and *Ae. tauschii*, 20 orthologous gene pairs were identified from *T. aestivum* and *T. dicoccoides*, 12 orthologous gene pairs were identified from *T. aestivum* and *T. urartu*, and 3 orthologous gene pairs were identified from *T. urartu* and *T. dicoccoides*. Given phylogenetic analyses and homology results of four wheat species, it was speculated that 8 *TaGRFs* (*TaGRF2-2B, TaGRF3-2B, TaGRF4-4A, TaGRF4-4B, TaGRF6-4A, TaGRF8-6B, TaGRF9-6A* and *TaGRF9-6B*) originated from *T. dicoccoides*, 9 TaGRFs (*TaGRF2-2D, TaGRF3-2D, TaGRF4-4D, TaGRF5-4D, TaGRF8-6D, TaGRF9-6D, TaGRF10-6D, TaGRF11-7D* and *TaGRF12-7D*) from *Ae. tauschii*, and 5 *TaGRFs* (*TaGRF3-2A, TaGRF8-6A, TaGRF8-6B, TaGRF10-6A* and *TaGRF12-7A*) from *T. urartu*.

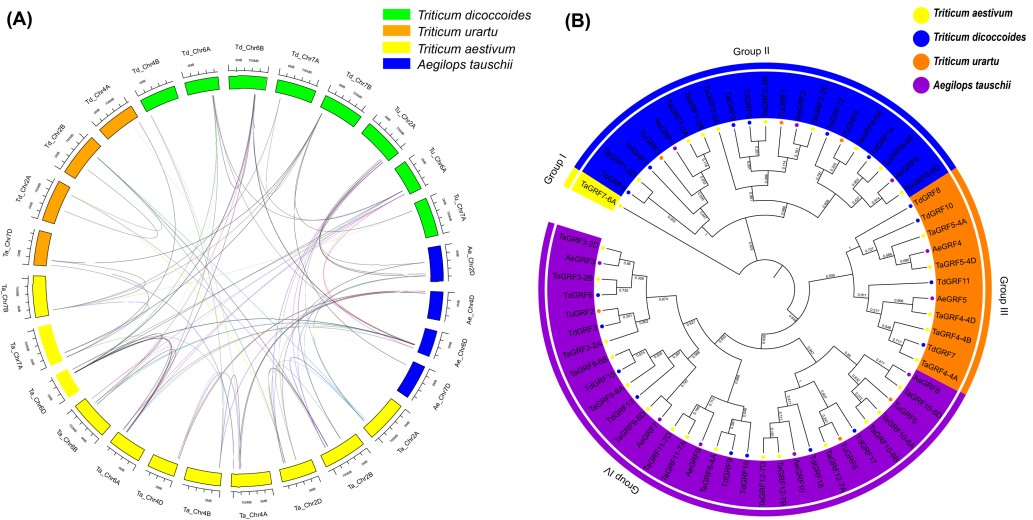

**Figure 7** Synteny (A) and Phylogenetic (B) analyses for GRF genes in T. aestivum and its subgenomic progenitors *T. urartu*, *T. dicoccoides* and *Ae. tauschii*. (A) Blue rectangles (Ae) represent *Ae. tauschii* chromosomes, Orange rectangles (Tu) represent *T. urartu* chromosomes, Yellow rectangles (Ta) represent *T. aestivum* chromosomes, Green rectangles (Td) represent *T. dicoccoides* chromosomes. (B) Phylogenetic relationship of *T. aestivum, T. urartu, T. dicoccoides*, and *Ae. tauschii*. This tree consists of 1,000 bootstraps created by the Neighbor-Joining (NJ) method in MEGA7. The blue, yellow, green and purple circles represent *T. dicoccoides*, *T. aestivum*, *Ae. tauschii*, *T. urartu*, respectively. All identified GRF genes are in corresponding chromosomes (see Table S2). The 89 homologous are shown in Table S7.

## Gene expression pattern analyses of *TaGRFs*

For multigene families, analysis of gene expression patterns often provides useful clues for determining gene function. Transcriptome data from growth and abiotic stresses were downloaded from Wheat Expression Browser to examine their expression patterns. The results showed that 28 *TaGRF* genes (except *TaGRF2-2A* and *TaGRF7-6A*) were expressed in different tissues or under different stress treatments (Fig. 8, and Table S8). *TaGRF1-2A* and *TaGRF1-2D* were highly expressed in various tissues. *TaGRF5-4A* and *TaGRF5-4D* had the highest expression in shoot tip meristem. Most of the genes were expressed in shoot tip meristems more significantly than other tissues. About half of the TaGRF genes were expressed under NaCl treatment, and *TaGRF4-4A*, *TaGRF4-4B* and *TaGRF4-4D* were significantly expressed under NaCl treatment. It was speculated that TaGRF family members may play important roles in the development of wheat shoot tip meristems, and *TaGRF4* may play an important role in wheat salt tolerance.

## Quantitative-real time PCR analysis

To further understand the potential role of the *TaGRF* genes in abiotic stresses (NaCl and mannitol), qRT-PCR was used to analyze the expression pattern of *TaGRFs*. Based on transcriptome analysis, we selected 14 *TaGRFs* for qRT-PCR. In the two treatments of this study, the expression of all 14 *TaGRFs* differed from the control, although their degree of difference was often substantial (Fig. 9).

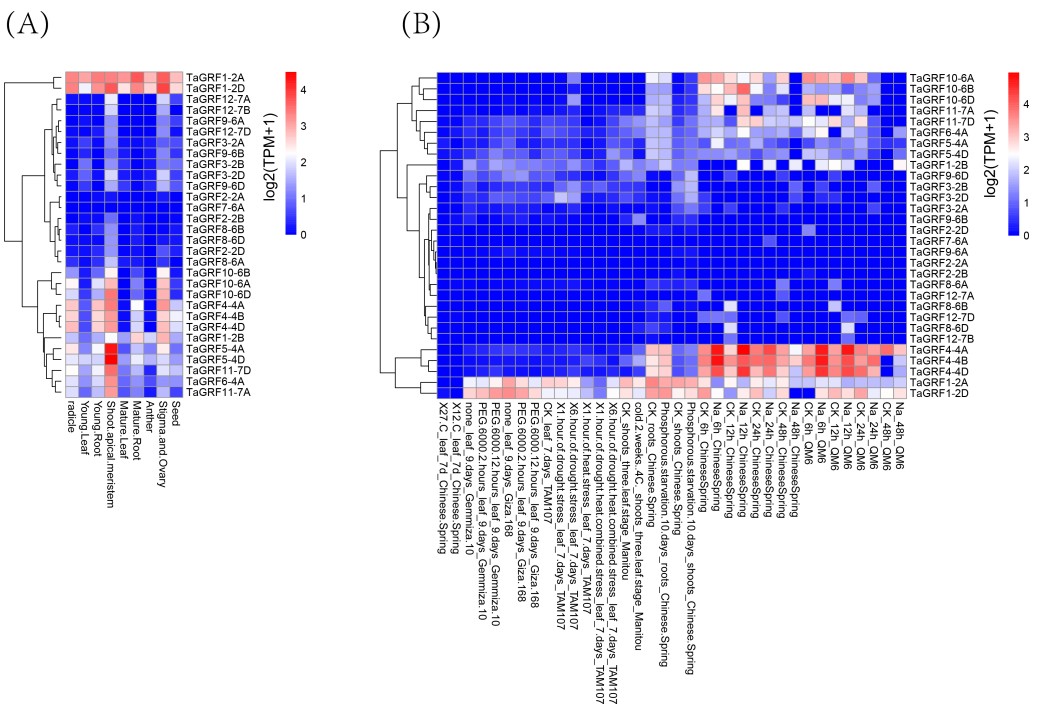

**Figure 8 Transcriptome analyses of 30 TaGRFs.** (A) Growth and development. (B) Abiotic stresses.

After treatment with NaCl, the expression of *TaGRF4-4A*, *TaGRF4-4D*, *TaGRF5-4A*, *TaGRF10-6A*, *TaGRF10-6B* and *TaGRF10-6D* were higher in treated leaves than in the control 96 h after treatment. Expression of the genes *TaGRF1-2B*, *TaGRF1-2D*, *TaGRF3-2D*, and *TaGRF9-6D* were lower in the leaves 96 h after treatment than in the control group. However, in the roots, the expression of *TaGRF1-2B* and *TaGRF3-2D* were much higher than in the control 96 h after treatment, while the expression of *TaGRF4-4A*, *TaGRF4-4D*, *TaGRF10-6B*, and *TaGRF10-6D* were much lower than in the control 96 h after treatment. Among these, the expression trends of *TaGRF1-2B*, *TaGRF3-2D*, *TaGRF4-4A*, *TaGRF4-4D*, *TaGRF10-6B*, and *TaGRF10-6D* were completely reversed in the roots compared to the leaves 96 h after treatment.

After treatment with mannitol, *TaGRF1-2B, TaGRF6-4A, TaGRF10-6A* and *TaGRF9-6D* were expressed higher in the roots than in the control group 24 h and 96 h after treatment. However, in the leaves, only *TaGRF6-4A, TaGRF10-6A* and *TaGRF9-6D* was expressed higher than the control group at 24 h and 96 h after treatment. The expression level of *TaGRF1-2B* did not change compared with the control group. *TaGRF1-2B* and *TaGRF3-2B* were significantly down-regulated in leaves at 2 and 4 h after treatment.

## DISCUSSION

With the in-depth development of plant genomics research, especially the rapid development of sequencing technology, the entire genome sequencing of many plant species has been completed, providing favorable conditions for the identification of plant

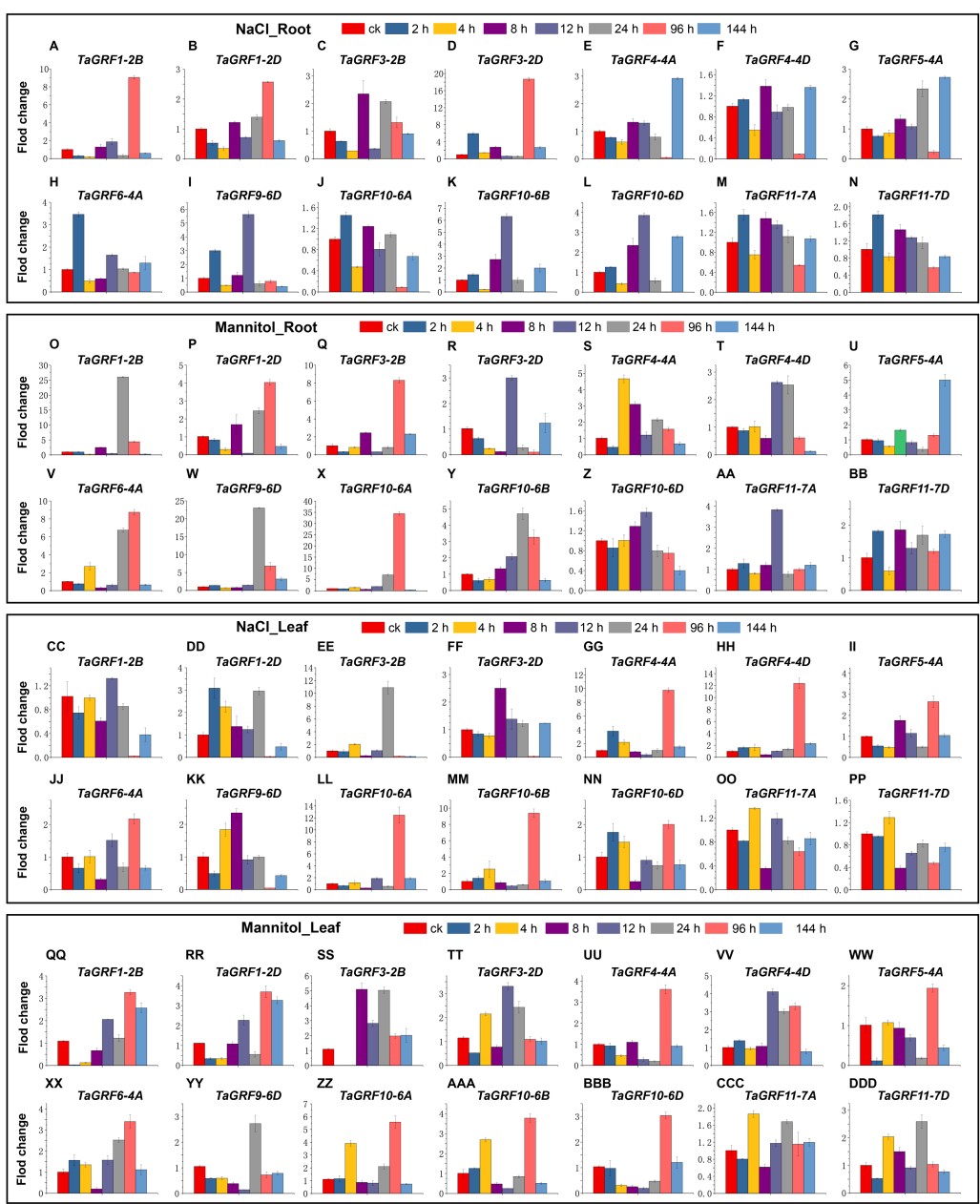

**Figure 9 The qRT-PCR analyses of 14 *TaGRFs* in roots and leaves after treatment with NaCl and mannitol.** (A-N) NaCl root (O-BB) Mannitol root, (CC-PP) NaCl leaf, (QQ-DDD) Mannitol leaf. Time periods shown on the *x*-axis. Expression levels are on the *y*-axis. Standard deviations are shown with error bars. The expression levels of TaGRF genes were plotted using Origin software.

gene families. GRF transcription factors are plant-specific transcription factors. In recent years, GRF transcription factor members of different plants have been identified and their genetic functions were studied. The results from these studies indicate that GRF genes are mainly expressed in plant meristems and play important roles in plant growth and development. In general, the number of GRF transcription factor members in terrestrial
plants is between 8 to 20, but, typically, fewer are found in lower plant taxa such as mosses and algae (*Omidbakhshfard et al., 2015*). For example, 9 AtGRFs occur in *Arabidopsis* (*Kim, Choi & Kende, 2003*), 12 OsGRFs in rice (*Choi, Kim & Kende, 2004*), 14 ZmGRFs in maize (*Zhang et al., 2008*), 17 BrGRFs in Chinese cabbage (*Wang et al., 2014*), 9 CsGRFs in sweet orange (*Citrus sinensis* L. Osbeck) (*Liu et al., 2016*), 18 PeGRFs in bamboo (*He et al., 2018*), while the moss (*Physcomitrella patens*) has only 2 GRFs (*Omidbakhshfard et al., 2015*). Based on wheat genomic data, the present study identified 30 wheat GRF transcription factors (TaGRF). Although the wheat genome is considerably larger than the *Arabidopsis* genome (16 GB vs 125 MB), the number of GRF genes in wheat is only three times that of *Arabidopsis* (30:9), indicating that there is a large amount of gene loss during genome replication in wheat. According to phylogenetic analysis, the 30 GRF transcription factors in wheat could be divided into four groups. Studies have shown that GRF transcription factors of rice and maize can be divided into three groups, five groups in *Arabidopsis*, and six in rapeseed, indicating that GRF transcription factors in monocotyledons different from dicotyledons in evolution patterns and characteristics. In this study, phylogenetic analysis of GRFs of wheat, *Arabidopsis*, rice, maize, *T. urartu*, *T. dicoccoides* and *Ae. tauschii* were carried out compared. It was shown that most TaGRFs preferentially clustered with GRF in *T. urartu*, *T. dicoccoides* and *Ae. taus* chii, followed by rice and maize. The results showed that the GRFs in wheat were closely related to those in *T. urartu*, *T. dicoccoides* and *Ae. tauschii*. The number of GRF genes in Group IV is greater than in Group I to III, implying that the variability in the number of GRF genes in the different groups may be the result of independent gene gain or loss in these groups. It is generally believed that exon-intron structure is important for understanding evolutionary and functional relationships (*Hu & Liu, 2011*). In addition, gain or loss events in exons or introns provide structural and functional differentiation (*Xu et al., 2012*). With regard to corresponding gene structures within each group, most *TaGRF* genes shared a similar gene structure, having two to four introns/exons, which is in accordance with *Arabidopsis* and rice (*Choi, Kim & Kende, 2004*; *Kim, Choi & Kende, 2003*). 22 of the *TaGRF* genes contained three or four exons and 29 of the *TaGRF* genes contained two or three introns. This indicated that, the structural evolution of the *TaGRF* gene is conservative to some extent.

Gene replication events are the main drivers of genome and genetic system evolution (*Moore & Purugganan, 2003*). Wheat has a complex evolutionary history with two major polyploid events (*Ling et al., 2013*). About 50–70 million years ago, before the genetic grouping of herbs, the first genome duplication directly produced an ancient doubling event. The second time was that the traceability of common wheat originated from the forming process of the tetraploid wheat (*T. dicoccoides*, A and B sub-genome) which hybridized by sub-genome progenitor *T. urartu* and *Aegilops speltoides* (B sub-genome) 300,000 years ago approximately. Again, about 8,000 years ago, the tetraploid wheat was hybridized with *Ae. tauschii* (D sub-genome) and formed hexaploid wheat (*T. aestivum*, A, B, and D sub-genome) naturally. We found that some genes were deleted during polyploidization by comparing GRF genes of *T. aestivum*, *T. urartu*, *Ae. tauschii* and *T. dicoccoides*.

Studies have shown that the expression level of the *GRF* gene is significantly higher in developing tissues than in mature tissues (*Kim, Choi & Kende, 2003*; *Kim & Kende, 2004*; *Choi, Kim & Kende, 2004*). For example, the *GRF* genes in rice were found to be strongly expressed in buds, immature leaves, and flower buds, and participates in plant growth and development by regulating cell proliferation in actively growing tissues (*Choi, Kim & Kende, 2004*). The expression profile of *TaGRF* genes analyzed by wheat tissue transcriptome data, showed that most of the *TaGRF* gene is highly expressed in wheat shoot tip meristems, and weakly expressed in other relatively mature tissues, which was similar to the previous conclusions. *OsGRF6* participates in regulating the growth and development of rice infloreses (*Gao et al., 2015*), and the three genes with the highest homology level, *TaGRF4-4A*, *TaGRF4-4B* and *TaGRF4-4D*, have high expression in stigma and ovary, indicating that their function may be related to the growth and development of stigma and ovary. In addition, cis-acting elements related to the regulation of meristem expression, such as cat-box and CCGTCC motif, were found in the promoter region of *TaGRF* genes, indicating that the *TaGRF* genes play important roles in wheat growth tissues, especially in stem tip meristems.

Plants have evolved a series of signal pathways and defense systems to resist stresses. In previous researches, the activation of genes responsing stresses enhanced the plant's tolerance (*Heidel et al., 2004*; *Sakuma et al., 2006*). Over-expression of *AtGRF7* in *Arabidopsis* under stress conditions increased resistance to osmotic and drought stress (*Kim et al., 2012*). It has been reported that GRF transcription factors acted as key roles in plant growth by coordinating stress responses and defense signals (*Casadevall et al., 2013*; *Liu et al., 2014a*; *Liu et al., 2014b*). For example, *Arabidopsis* growth regulators 1 and 3 *(AtGRF1* and *AtGRF3)* played significant roles in the regulation of plant growth, defense signals, and stress responses (*Casati, 2013*; *Hewezi et al., 2012*). In our study, the *cis*-elements of 12 *TaGRFs* (*TaGRF1-2A*, *TaGRF1-2B*, *TaGRF1-2D*, *TaGRF3-2D*, *TaGRF4-4A*, *TaGRF5-4A*, *TaGRF6-4A*, *TaGRF9-6A*, *TaGRF9-6B*, *TaGRF9-6D*, *TaGRF12-7A*, and *TaGRF12-7D*) contained 1 to 2 copies of MBS (the MYB binding site is involved in drought-inducing). The qRT-PCR results showed that under NaCl stress and mannitol simulated drought stress, 14 *TaGRF* genes we tested responded to external abiotic stresses, either positively or negatively. Among them, 9 genes (*TaGRF1-2B*, *TaGRF3-2B*, *TaGRF3-2D*, *TaGRF4-4A*, *TaGRF4-4D*, *TaGRF6-4A*, *TaGRF9-6D*, *TaGRF10-6A*, *TaGRF10-6B*) were significantly expressed in treatment with NaCl and mannitol. These genes may play an active role in wheat's response to NaCl stress and drought stress. According to transcriptomic data and qRT-PCR results, *TaGRF1-2D*, *TaGRF4-4A* and *TaGRF4-4D* were all up-regulated in salt stress, indicating that they may play a certain role in wheat response to salt stress. But more experimental evidence is needed to understand how they work in wheat in response to salt stress.

## CONCLUSIONS

This study provides a reference point for subsequent studies involving functions of the *TaGRF* gene family. *TaGRF* gene family has extensive expression profiles which

span multiple developmental stages and stresses, implying their crucial roles in various physiological functions and abiotic stresses. In summary, our findings provide new clues that will be useful for improving stress tolerance of wheat.

### Funding
This study was funded by the Key Projcet of Hubei Provience Departmen of Education (grant number D20191305). The funders had no role in study design, data collection and analysis, decision to publish, or preparation of the manuscript.

### Grant Disclosures
The following grant information was disclosed by the authors:
Hubei Provience Departmen of Education: D20191305.

### Competing Interests
The authors declare there are no competing interests.

### Author Contributions
- Wendi Huang and Cai Sun analyzed the data, prepared figures and/or tables, authored or reviewed drafts of the paper, and approved the final draft.
- Yiqin He and Lei Yang performed the experiments, prepared figures and/or tables, and approved the final draft.
- Chen Lu performed the experiments, authored or reviewed drafts of the paper, and approved the final draft.
- Yongxing Zhu analyzed the data, authored or reviewed drafts of the paper, and approved the final draft.
- Dongfang Ma conceived and designed the experiments, authored or reviewed drafts of the paper, and approved the final draft.
- Junliang Yin conceived and designed the experiments, prepared figures and/or tables, authored or reviewed drafts of the paper, and approved the final draft.

### Data Availability
Raw data are available in the Supplemental Files.

### Supplemental Information
Supplemental information for this article can be found online at http://dx.doi.org/10.7717/peerj.10701#supplemental-information.

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
