# Peer review of "Genome-wide analysis of growth-regulating factors (GRFs) in Triticum aestivum"

_PeerJ, doi:10.7717/peerj.10701_

## Round 0.1 · original submission · Major Revisions

Both Reviewers have several suggestions for improvement of the manuscript. Please, follow their suggestions and resubmit a revised manuscript. Please, add a letter describing all changes in the revised manuscript based on the reviewers' comments. Please, pay attention to general language improvement.

Reviewer 1 ·

Basic reporting

In this reasonably well written manuscript, Huang et al. describe the identification and bioinformatics analysis of the GRF family in Triticum aestivum. In addition, they analyzed their expression patterns under drought and salt stress, which will facilitate further study on the relationship between GRF gene family in Triticum aestivum and its development regulation network and stress response. They have used clear and professional English throughout the paper.

Major issues:
1.Figure 3: In the legend to the Figure, the author declares that the neighbor-joining method is used, but in the manuscripts, he writes Maximum Likelihood method (Line 103). The two are inconsistent. The author needs to clearly use the method, and the phylogenetic relationships generated by different methods may differ significantly.

2.Figure 3: The author needs to be in the analysis of evolution bootstraps values marked in the Figure, high bootstraps values on the branch gene deserves more attention, at the same time can also be found TaGRF genetic relationship with the distance of GRF gene in different species.

3. Figure 3 and Figure 1: In the two Figures, the author claimed that both phylogenetic relationships were constructed by neighbor-joining method, but the results of the two Figures were inconsistent. Assuming that GRF genes of other species in Figure 3 were excluded, the evolutionary relationship of TaGRF in Figure 3 was inconsistent with that in Figure 1.

4.Line 286-289: The authors speculate that the conclusion that TaGRF family plays an important role in wheat shoot tip meristems and salt stress is not rigorous enough. In Figure 7, only a few TaGRF genes show significant changes in expression under the above conditions, which is not enough to indicate that the whole GRF gene family has the same role.

5.Figure 7 and Figure8: The author should link some important genes in the two Figures, such as TaGRF1-2D, TaGRF4-4A and TaGRF4-4D. The above-mentioned genes play an important role in the tissue expression pattern and salt stress in FIG. 7, but unfortunately the author did not correlate them with the transcriptome data in FIG. 8, and further explain the problem.

Experimental design

1.The authors should use bioinformatics methods to identify several highly reliable TaGRF genes, otherwise the experiment will not be accurate and definitive conclusions.

Validity of the findings

No.

Additional comments

minor issues:
1.Line 97-99: It is suggested that the authors submit the protein sequences of candidate genes to the NCBI CDD (https://www.ncbi.nlm.nih.gov/Structure/cdd/wrpsb.cgi) and SMART (http://smart.embl-heidelberg.de/smart/set_mode.cgi?NORMAL=1) databases for the presence of the conserved domain of GRF.

2. The author needs to explicitly refer to the source of the software used, such as Line 134, TBtools software.

3. Figure 1:It is recommended that the authors rename motif 1 and 2 to WRC and QLQ to be more intuitive.



4.Line 89: of and Pfam lack of space.

5. Some punctuation is not standard and English spelling problems.

·

Basic reporting

no comment

Experimental design

no comment

Validity of the findings

no comment

Additional comments

In this study, the wheat GRF gene family was identified and comprehensively analyzed including phylogenetic evolution, gene structure, and gene expression.
This has not been reported in wheat, which will provide a theoretical basis for the study of plant development and stress. The overall experimental methods and strategies are correct, and the experimental content is complete. However, the structure and logic of the entire manuscript need to be improved, and the expression should be more professional.

Introduction part:
Abbreviation should be interpreted where it first appears, (Gln, Leu, Gln) and (Trp, Arg, Cys) should be placed in line 41 instead of lines 42 and 44.


Lines 51-53 need inferences.


I think the functions and mechanisms of GRF participation should be listed in more detail and more organized. In lines 51-61, you explained GFPs' participation in the formation of plant tissues or organs, but there was no turning point or summary when you cited other functions of GFP in lines 62-66. In order to laying the groundwork for your research, you should describe its possible mechanisms in more detail rather than simply describe the functions in defense responses (lines 63-66). Also, pay attention to English expressions.


GRF TFs not only participate in plant growth and development, but also respond to certain abiotic stresses (Kim et al., 2012).
It is more appropriate to put this sentence in the previous paragraph than here.


It is better to put a separate paragraph on the brief introduction of content of this study (lines 76-82) and it should briefly introduce the significance and value of this research.

Material and Methods
The e-value of hmmer search should be given (line90).


In lines 97-99, how to eliminate sequences which did not belong to the GRFs? Exclude sequences that do not contain both WRC and QLQ, or sequences that contain one of the domains are also excluded?

In lines 112-117, the evaluation criteria for duplication events are strict and reasonable, but here you need to add what strategies and software were used to obtain tandem and segmental duplications.


Lines 141-145, The SRA ID of the RNA-seq data should be given, the version and references of the software used should be given.

Four Species ? The orthologous genes from T. urartu, T. dicoccoides, and Ae. tauschii were identified by reciprocal BLASTp analysis.

There are no problems with the method used in this study, but there are two major deficiencies in the presentation in this section. (a) The content arrangement is a bit messy and should correspond to the result part. For example, ‘Characterization Prediction of TaGRF Proteins’ was placed in the fourth paragraph, but in fact it was described before you described duplication events in the results section. Similar problems need to be modified. Individual paragraph content can be merged into one paragraph. (b) Language description should pay attention to avoid Chinglish. Sentence-by-sentence description of some content will be ambiguous, you can merge two sentences into one by using clauses, etc. E.g, lines 130-131. In addition, the subtitle expression has grammatical problems, especially the following three.
Sequence Alignment and Construction of Phylogenetic Tree
TaGRFs Expression Profiling by Mining Multiple Transcriptome Data
Wheat Seedlings Growth and Stress Treatments


Lines 184-186, what is the result? How many TaGRF you identified ? In addition, you used two strategies instead of only blastP search.

Line 186, TaGRF should be shown in Italics when it present gene.

Line 197, former grouping rules, what does it mean? References should be given.

Line 210-211, to supplement the method of multiple sequence alignment in material and methods part.

It’s reasonable to put phylogenetic analysis before structure analysis. Otherwise, how to explain the basis of the evolutionary tree in Figure 1? In fig 2, it is better to show phylogenetic tree in a suitable appearance for easier to find their corresponding names for some short branches.

It’s reasonable to put chromosome localization analysis after identification of TaGRFs, because you renamed them according to chromosome localization. It is also not very appropriate for two sentences to form a paragraph. Don’t show the duplication events in fig5.


Lines 250-262, this content should be put in discussion part.


Line 249-277, this content should be followed after identification and phylogenetic analysis (with Arabidopsis, rice, and maize). The subtitle is inappropriate and the results should be properly detailed.

Discussion
The discussion section should be carefully revised.
This part should be combined with the results of this study and existing research for discussion, rather than use a super paragraph to describe the experimental results in isolation, and another paragraph to introduce future research.


Figures and Table
The legends of fig 4, fig 5, fig 6 and fig 8 should be more detailed.

In this study, the related information of signal peptide should be removed because no result have been obtained.

---

## Round 0.2 · Minor Revisions

There are still some concerns regarding the citations by one Reviewer.

Furthermore the Editor has following comments and editorial improvements:

line 322:...four groups..
line 323: ...three groups, five groups....
line 327:... Ae. tauschii….
lines 360,361: ….stigma and ovary…..
line 368: It has been reported…..acted as….
line 380: ...wheat's….

line 254: the sentence:... A total of… is incomprehensive and wrong in grammar. Revise!
line 264: Correct….TaGRF1-22B

Lines 262-269:
I am wondering how genes on the D-genome can be derived of T. dic., genes on the A- and B-genomes from Ae. tauschii and genes on the B- and D-genomes from T. urartu.

For comparison to international literature I suggest to mention in some table or Supplement the relationship between your TaGRF gene IDs and the official gene IDs of the reference sequence (Traes…)

Please, provide a revised manuscript with accompanying letter describing the revisions.

Reviewer 1 ·

Basic reporting

this revised MS addressed most of my concerns.

Experimental design

this revised MS addressed most of my concerns.

Validity of the findings

this revised MS addressed most of my concerns.

Additional comments

this revised MS addressed most of my concerns.

·

Basic reporting

The author has solved all the comments except the marking format and document format of the literature. With minor modifications, the paper can be accepted.

Experimental design

no comment

Validity of the findings

no comment

Additional comments

Although the author has revised it according to the requirements, There are many mistakes in the quotation and description format of the literature.
1. Line 39, ...‘.in regulating gibberellic acid (GA)-induced stem elongation (Knaap, Kim & Kende, 2000).Line 43, in the N-terminal region of the GRF protein (Kim, Choi & Kende, 2003)......’should be ...‘.in regulating gibberellic acid (GA)-induced stem elongation (Knaap et al., 2000).Line 43, in the N-terminal region of the GRF protein (Kim et al., 2003)......’ etc. Line 49, 'he C-terminal region has trans-activation activity (Kim & Kende, 2004; Choi, Kim & Kende, 2004; Liu et al., 2014).' should be ...‘.the C-terminal region has trans-activation activity (Choi et al., 2004; Kim et al. 2004; Liu et al., 2014). ....’ etc.
2. The full name or abbreviation of journals involved in literature should be unified.

---

## Round 0.3 · Major Revisions

I am a Section Editor for Plant Biology and I am taking over from the previous Academic Editor for this decision.

Upon further review, there appear to be significant revisions which would greatly help this manuscript, as it does not distinguish itself from prior data available from genome resources, and as it presents a strong point for comparisons between species, additional information should point or be made available for the readers to be guided to the orthologous regions.

Table 1 is not complete (cut in half vertically). If there are new descriptors the updated GFF file annotations should be provided as they can be added to the previous works. If authors rely on previous works they should be willing to contribute back with updated resources. As there is a pool of sequences being highlighted it is important to discern them between the four species highlighted and to point out which are new annotations versus the previous, and as the reference versions were highlighted; the GFF files may be the best way to describe them as figures to little to point out the motifs mentioned. These can be added as supplements but should hold the new annotations which distinguish it from the original reference source.

The information does appear valuable, but it should be made clear that the annotations are different from previous annotations for release 1.1 of the Triticum genome, and that new annotations can be generated for the other species.

While there was some effort to refine GRF annotations there is no apparent effort to refine the expression based on ontology terms since it does appear that biological, molecular, and cellular expression is a goal for this effort. As this appears more a bioinformatics exercise with little additional expression work the additional efforts to improve annotations would appear warranted. I rank this as requiring additional revision.

In addition. I would suggest a change of title to:
"Genome-wide analysis of growth-regulating factors (GRFs) in Triticum aestivum"

Apologies for the delays. The needed revision may already be in hand based on the descriptions within the manuscript.

---

## Round 0.4 · Minor Revisions

Thank you for the edits which were performed; they filled in many of the requests, but not all. I am somewhat moved to recommend an accept decision for the manuscript, but there may be some difficulty for comprehension in some areas. I note that Table S2 highlights the species areas which are being described; however, this information may not be present in Table S3, as it only shows Triticum aestivum coordinates. Perhaps this can be corrected, but at least the reader can try to re-create the connection from the information in Table S2, with additional effort. You may want to have the manuscript properly reviewed by a language proofing service. This is not my role, but provided I have time I will try to further review the context of the manuscript. In the meantime, improving the Table S3 would be a help.

---

## Round 0.5 · accepted · Accept

Thank you for the edits which were performed; it appears all that was asked for was attended to. I believe the manuscript is now ready to move forward. I will tag this as ready for publication. The manuscript appears to demonstrate the wide range of GRF gene expression profiles, and provides a vehicle for additional comparative studies to be performed. Apologies for all the delays, but I believe the suggested edits helped improve the presentation of the work. Thank you for your efforts.